# Women's perceptions and experiences of reproductive coercion and abuse: a qualitative evidence synthesis

**Jessica E. Moulton**⊙*, **Martha Isela Vazquez Corona**⊙, **Cathy Vaughan, Meghan A. Bohren**

Gender and Women's Health Unit, Centre for Health Equity, School of Population and Global Health, University of Melbourne, Carlton, VIC, Australia

* jessica.moulton@monash.edu

**Data Availability Statement:** All relevant data are within the paper and its S1 File, S1 Checklist, S1–S5 Tables files.

## Abstract

### Background

Reproductive coercion and abuse is a major public health issue, with significant effects on the health and well-being of women. Reproductive coercion and abuse includes any form of behaviour that intentionally controls another person's reproductive choices. The aim of this qualitative evidence synthesis is to explore women's experiences of reproductive coercion and abuse globally, to broaden understanding of the different ways reproductive coercion and abuse is perpetrated, perceived and experienced across settings and socio-cultural contexts.

### Method

We searched Medline, CINAHL and Embase for eligible studies from inception to 25th February 2021. Primary studies with a qualitative study design that focused on the experiences and perceptions of women who have encountered reproductive coercion and abuse were eligible for inclusion. Titles and abstracts, and full texts were screened by independent reviewers. We extracted data from included studies using a form designed for this synthesis and assessed methodological limitations using CASP. We used Thomas and Harden's thematic analysis approach to analyse and synthesise the evidence, and the GRADE-CERQual approach to assess confidence in review findings.

### Results

We included 33 studies from twelve countries in South Asia, the Asia Pacific, North America, South America, Africa and Europe. Most studies used in-depth interviews and focus group discussions to discuss women's experiences of reproductive coercion and abuse. Reproductive coercion and abuse manifested in a range of behaviours including control of pregnancy outcome, pregnancy pressure or contraceptive sabotage. There were a range of reasons cited for reproductive coercion and abuse, including control of women, rigid gender roles, social inequalities and family pressure. Women's different responses to reproductive

**Funding:** The author(s) received no specific funding for this work, which was completed as part of JEM's MPH at University of Melbourne School of Population and Global Health. MAB's time is supported by an Australian Research Council Discovery Early Career Researcher Award (DE200100264) and a Dame Kate Campbell Fellowship (University of Melbourne Faculty of Medicine, Dentistry, and Health Sciences).

**Competing interests:** The authors declare that they have no competing interests.

**Abbreviations:** IPV, Intimate Partner Violence; USA, United States of America.

coercion and abuse included using covert contraception and feelings of distress, anger and trauma. Across contexts, perpetration and experiences of reproductive coercion and abuse were influenced by different factors including son preferences and social exclusion.

## Conclusions

We reflect on the importance of socio-cultural factors in understanding the phenomenon of reproductive coercion and abuse and how it affects women, as well as how the mechanisms of power and control at both individual and societal levels work to perpetuate the incidence of reproductive coercion and abuse against women.

## 1 | Background

Reproductive coercion and abuse is a major public health issue, with significant effects on the mental, sexual, reproductive and maternal health of women who have experienced it. Reproductive coercion and abuse includes any form of behaviour that intentionally controls another person's reproductive choices [1]. These behaviours include forcing a person to continue or terminate a pregnancy, or sabotaging contraception, for example, by removing or damaging a condom, or throwing away oral contraceptives [2]. Reproductive coercion and abuse is often a manifestation of a partner's demand to enforce their own reproductive intentions [3], using physical, psychological, sexual, financial and other strategies with the purpose of maintaining power and control within the relationship [4]. These threats and acts of violence often overrule a woman's ability to exercise their reproductive rights and autonomy [5].

Reproductive coercion and abuse is a relatively recent term used to denote a pattern of behaviours described in the gender-based violence literature. The term 'reproductive coercion and abuse' was first mentioned in academic literature in 2010 [6]. However, well before this, pregnancy-controlling behaviours by male partners have been described in the gender-based violence literature without being labelled as reproductive coercion and abuse [7]. Prevalence rates of reproductive coercion and abuse have ranged from 8.6% of all women in The National Intimate Partner and Sexual Violence Survey in the United States of America (USA) to as high as 37.8% among young, self-identified Black or African American women in Baltimore, Maryland [8]. However, a lack of consistent measurement and conceptual clarity around reproductive coercion and abuse mean that true prevalence rates are inconclusive [9]. A systematic review by Grace and Anderson [7] found that in the USA, reproductive coercion and abuse disproportionately affects women experiencing other forms of intimate partner violence, women of low socioeconomic status and women who are Latina, African American, or multiracial. Similarly, in an Australian study of prevalence among women using a pregnancy counselling service, Indigenous and culturally-and-linguistically-diverse women were more likely to have experienced reproductive coercion and abuse [10]. Reproductive coercion and abuse also disproportionately affects younger women, regardless of whether other forms of intimate partner violence are present [6, 11, 12]. Research has shown that women who have experienced intimate partner violence and controlling or coercive behaviour are more likely to experience contraception sabotage or unintended pregnancy [6, 7]. In a 2010 study of 1200 sexually active women (16-29 years), over one-third of women who reported experiencing intimate partner violence had also experienced reproductive coercion and abuse [6]. Likewise, women in Australia who experience domestic violence are twice as likely to report behaviours such as condom refusal and unplanned pregnancy [10].

Reproductive coercion and abuse can be explained in a social structure where cultural and social norms of male dominance over women are widely upheld [13]. Men's assertion of power and control of women through abusive tactics has been researched extensively. Heise's highly influential ecological framework can be used for understanding the causes of gender-based violence and to conceptualise reproductive coercion and abuse as an interplay between personal, situational and sociocultural factors [13]. Factors that may influence the likelihood of perpetration of reproductive coercion and abuse include personal factors such as perpetrators having experienced abuse themselves or history of abusive behaviour, interpersonal factors such as friends or family members of the perpetrator who are supportive of violence and situational and sociocultural factors such as high rates of unemployment, religion, and policy and social and cultural norms in which violence against women is permissible [13].

Women of different nationalities, cultures and social customs, with varying degrees of legal rights, may experience reproductive coercion and abuse differently [1]. Factors that may increase risk in some countries may, conversely, be protective in other settings [14]. A qualitative evidence synthesis exploring the experiences of women globally will broaden understanding of the different ways reproductive coercion and abuse is perpetrated, experienced and perceived across different socio-cultural contexts. Understanding the experiences and perspectives of women who have encountered these behaviours will provide greater clarity of the range of experiences, responses, and impacts of reproductive coercion and abuse, and provide valuable insight for developing prevention, screening and response programs. The aim of this qualitative evidence synthesis is to 1) identify the perceptions and experiences of women who have encountered reproductive coercion and abuse; and 2) identify barriers that women face to receiving appropriate interventions.

## 2 | Methods

We report our qualitative evidence synthesis based on the Cochrane Effective Practice and Organisation of Care template for qualitative evidence synthesis [15] and ENTREQ statement for enhancing transparency in reporting the synthesis of qualitative research [16] (S1 Table). The review protocol was not registered, and was conducted as part of JEM's MPH at University of Melbourne.

### 2.1 | Criteria for considering studies for this synthesis

**2.1.1 | Types of studies.** We included primary studies with a qualitative study design (e.g. ethnography, phenomenology) that used qualitative methods for data collection (e.g. focus group discussions, observation, individual interviews) and data analysis (e.g. thematic analysis, grounded theory). Any studies that used qualitative methods for data collection, but were not analysed qualitatively, were excluded. Mixed methods studies were included where it was feasible to extract data that was both collected and analysed with qualitative methods. Studies were not excluded based on the evaluation of methodological limitations; however, this information was used to measure our confidence in the review findings [17].

**2.1.2 | Topic of interest.** Reproductive coercion and abuse was broadly conceptualised as contraceptive sabotage, pregnancy coercion and control of pregnancy outcomes as defined in the Marie Stopes '*Hidden Forces*' White Paper on Reproductive Coercion [2]. Behaviours that constitute reproductive coercion and abuse include 1) sabotaging another's contraception; 2) controlling a pregnancy outcome (forcing someone to have a termination or continue a pregnancy); 3) forcing or pressuring someone into pregnancy and 4) forcing someone to have a sterilisation procedure [2].

Studies of reproductive coercion and abuse perpetrated by an intimate partner, family member, or in-law were included. Studies that only explored intimate partner or sexual violence, or reproductive coercion and abuse perpetrated by the state through policy and legislation were excluded (e.g. forced sterilisation of women by the state). As posited in a conceptual re-evaluation of the term reproductive coercion and abuse [9], when exploring the experiences of reproductive coercion and abuse, only behaviours where there is a perceived explicit intent of either impregnating a woman, preventing her from becoming pregnant or ending a pregnancy will be considered.

**2.1.3 | Types of participants and settings.**   Studies that focus on the experiences and perceptions of women who have encountered reproductive coercion and abuse were included (defined by themselves or the researchers). Studies from any country were included.

## 2.2 | Search methods for identification of studies

We searched the following electronic databases for eligible studies from inception to 25 February 2021: MEDLINE, Embase, CINAHL (S2 Table: Search strategies). Search strategies for each database were developed using the Cochrane Qualitative Research Methods Group guidelines for searching for qualitative evidence [18] and input from an information specialist. There were no search limitations based on language, date or geographic location. The reference lists of included studies were reviewed to identify other relevant references.

## 2.3 | Data collection, management and synthesis

**2.3.1 | Selection of studies.**   Citations identified through database searches were compiled into EndNote, where duplicates were removed, then uploaded to Covidence [19], where further duplicates were removed. All titles and abstracts of the identified studies were assessed independently by two reviewers (JEM, MVC) to evaluate eligibility for inclusion. We retrieved the full text of potentially relevant records and assessed them independently by two authors (JEM, MVC) for inclusion eligibility [17]. Any conflicts were resolved through consensus or discussion with a third author (MAB) [20]. Studies that were not published in English were listed as 'studies awaiting classification', to ensure that the review process remained transparent [17]. These studies were not included in the analysis.

**2.3.2 | Data extraction.**   Data were extracted from the selected studies using a Word template designed specifically for this review by JEM and checked by a second reviewer (MVC, MAB). The template included information about the study setting, sample characteristics, objectives, design, data collection and analysis methods, qualitative findings, supporting quotations, conclusions as well as any relevant tables, figures or images [17].

**2.3.3 | Assessment of the methodological limitations in included studies.**   An adaptation of the CASP tool (www.casp-uk.net) was used to assess methodological limitations for each included study (JEM, MVC, MAB) [17]. Methodological limitations were assessed according to the following domains: aims, methodology, research design, recruitment strategy, data collection, author reflexivity, ethical considerations, data analysis, statement of findings and research contribution. We did not conduct an overall assessment [15] rather, the concerns regarding each of these domains is described in detail (S3 Table). The Characteristics of Included Studies table (S4 Table) presents an overview of the studies and the assessment of methodological limitations for all studies.

**2.3.4 | Management and synthesis.**   A thematic synthesis approach described by Thomas and Harden [21] was used to analyse and synthesise the data. Thematic synthesis is used to analyse qualitative data by generating meaning from people's perspectives and experiences. We used a three-stage approach to analysis. First we conducted free line-by-line coding of the

findings in primary studies, then organised free codes into common themes and developed 'descriptive' themes. Finally, these themes were constructed into 'analytical themes', which were then interpreted to generate wider concepts and hypotheses [21]. Five articles most relevant to the review question were selected as the base for developing the code list. These initial codes were considered 'free codes' with no links developed. The codes were cross-checked to ascertain whether the concepts are transferrable between studies. From this the codebook was developed with new codes added as required (see S5 Table). Codes were compared and contrasted, and a hierarchical structure was utilised to group the codes. If new codes arose, studies were re-examined to determine whether they applied. All studies were then coded to form analytical themes that were more explanatory in nature [22]. Qualitative analysis was conducted using NVivo (QSR International Pty Ltd. Version 12, 2018).

**2.3.5 | Assessment of confidence in the synthesis findings.**  JEM, MVC and MAB used the GRADE-CERQual (Confidence in the Evidence from Reviews of Qualitative research) approach to assess the confidence each review finding [23]. CERQual uses the following four components to assesses confidence in the findings: **methodological limitations of included studies** [24]**, coherence of the review finding** [25], a**dequacy of the data** contributing to a review finding [26], and r**elevance of the included studies to the review question** [27]: After each of these components was assessed, our overall confidence in the evidence supporting the review finding was rated as high, moderate, low or very low [28]. Confidence assessments started at high confidence, and if concerns were found then confidence was downgraded [28].

# 3 | Results

Of the 9151 articles screened, 33 studies were included in the evidence synthesis (see Fig 1: Study flow diagram). All studies included the perspectives of women, four studies were mixed methods and 29 were qualitative studies. Of the 33 studies, 16 were from USA [3, 29–43], four from Kenya [44–47], three from Australia [48–50], two from India [51, 52], and one each from Brazil [53], Fiji [54], Iran [55], England [56], Sweden [57], Mexico [58], Ecuador [59] and Canada [5]. Nine studies specifically explored experiences of low-income women [30, 33, 36, 37, 40, 41, 43, 51, 52]. Two studies explored the experiences of Indigenous women in Australia [48, 49], two studies explored perspectives of rural Hindu women in India [51, 52], and three explored experiences of migrant women in Australia and USA [38, 42, 48]. Fifteen studies from the USA explored the experiences of women from different racial and ethnic backgrounds, particularly White, African American and Latina women [3, 29–37, 39, 40, 42, 43]. All women in the studies experienced reproductive coercion and abuse from male partners.

Reproductive coercion and abuse was the primary phenomena of interest in 13 studies [3, 5, 29, 40–42, 44, 47, 48, 50, 54, 58]. Reproductive coercion and abuse was a secondary phenomena of interest in the other studies, which focused on women's experiences with intimate partner violence (IPV) particularly sexual violence [31, 53, 57], the effects of abuse or IPV on women's reproduction [32–35, 39, 43, 45, 46, 51, 55, 56], factors affecting women's pregnancy intentions [30, 36, 49, 52], barriers to contraception use [37] and son preferences [38].

Thirty two studies described women's experiences of reproductive coercion and abuse within the three domains of pregnancy coercion, contraceptive control or control of pregnancy outcome [3, 5, 29–59]. Eighteen studies included perceived and experienced consequences of reproductive coercion and abuse [3, 5, 29, 32–36, 38, 39, 41, 43, 47–50, 56, 59] and 16 studies explored women's responses [3, 29, 30, 32, 33, 35–37, 42, 44, 45–49, 51].

Thirteen studies explored societal and cultural factors [3, 29, 33, 38, 42–44, 46, 48, 50–52, 55] and eight studies described men's contradictory behaviours after perpetrating reproductive coercion and abuse [3, 5, 29, 34, 35, 40, 56, 58]. Most studies focused on reproductive coercion

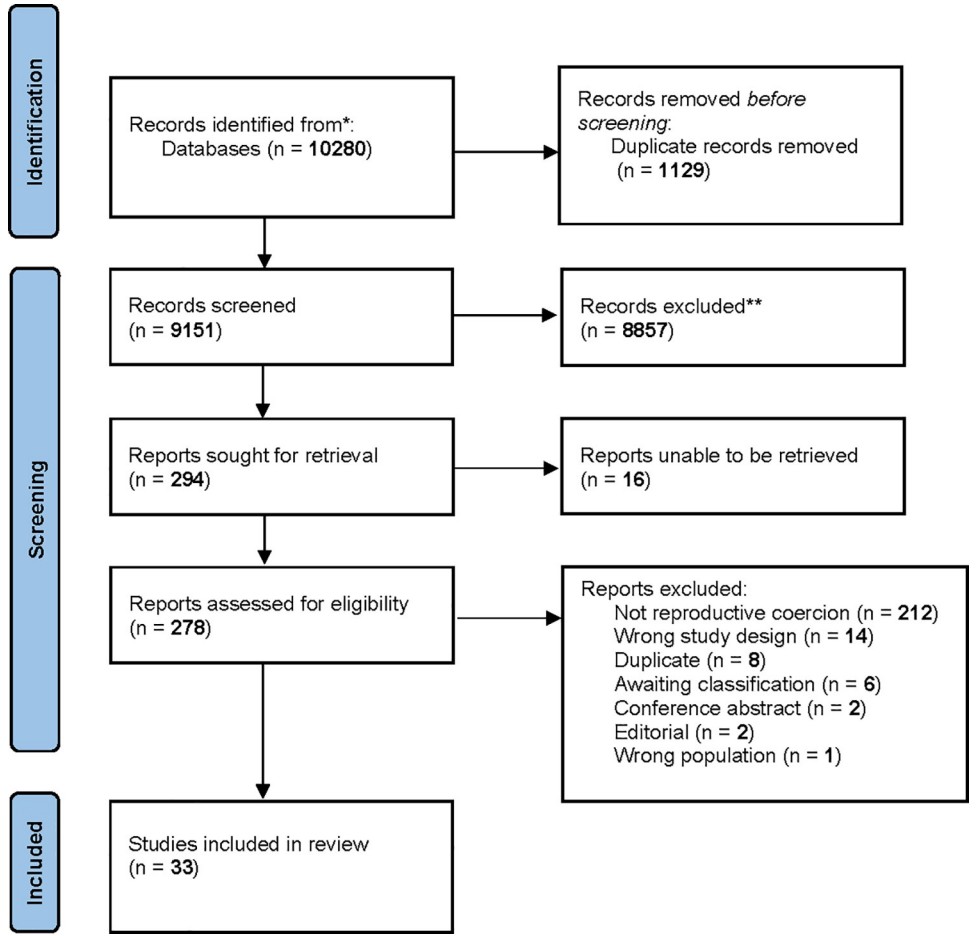

**Fig 1. Study flow diagram.** This figure depicts the study inclusions and exclusions.

and abuse by an intimate partner, and 10 studies described reproductive coercion and abuse perpetrated by family members or in-laws sometimes in conjunction with intimate partners [33, 38, 40, 43, 44, 46, 50–52, 55].

Table 1 presents a summary of qualitative findings and CERQual assessments. The following sections report the findings organised under the major themes: 1) women's experiences of reproductive coercion and abuse, 2) consequences of reproductive coercion and abuse and abuse, 3) women's responses to reproductive coercion and abuse and abuse, and 4) societal and cultural factors.The authors have also developed a logic model to organise and depict review findings (see Fig 2: Reproductive coercion and abuse: from causes to consequences).

## 3.1 | Women's experiences of reproductive coercion and abuse

This section outlines the different forms of reproductive coercion and abuse experienced by women. This includes contraceptive sabotage, forced sterilisation, emotional or verbal coercion and sexual violence. 32 studies explored women's experiences of reproductive coercion and abuse [3, 5, 29–59].

**Finding 1**

**Women experienced diverse forms of contraception sabotage by their partners, in order to assert control over them and obtain a desired reproductive outcome. Contraceptive**

**Table 1. Summary of qualitative findings.**

| # | Summary of review finding | Studies contributing to the review finding | CERQual assessment | Explanation of CERQual assessment |
|---|---|---|---|---|
| *Women's experiences of reproductive coercion and abuse* | | | | |
| 1 | Women experienced diverse forms of contraception sabotage by their partners, in order to assert control over them and obtain a desired reproductive outcome. Contraceptive sabotage included disposing of, withholding, or blocking access to contraception, interfering with condom durability and condom refusal as well as psychological sabotage in the form of deceit regarding male infertility, misinformation and gaslighting to interfere with contraception | [3, 29–31, 33–37, 42, 44, 47–49, 56–59] | High confidence | No to very minor concerns regarding coherence and adequacy and minor concerns regarding methodological limitations [reflexivity, ethics, data analysis, methodology, recruitment], and relevance [most studies conducted in the USA] |
| 2 | Some women were forced by their partners into permanent methods of contraception, such as tubal ligation | [33, 34] | Low confidence | Minor concerns regarding coherence and methodological limitations [aims, reflexivity, findings supported] and serious concerns regarding relevance and adequacy [2 studies with thin data conducted in the USA] |
| 3 | Partners used a range of verbally and emotionally coercive behaviours, such as harassment, pressure and bullying, to promote pregnancies that were unwanted by women. Partners used emotional manipulation, including threatening to end the relationship, if women did not get pregnant, or telling family and friends that they were starting a family without her knowledge. Some women experienced sexual violence by their partners, with the intention to cause pregnancy, including rape and non-consensual unprotected sex | [3, 29, 32–36, 38, 40, 41, 44, 48, 49, 51, 52, 56, 57] | High confidence | No to very minor concerns regarding coherence and adequacy, minor concerns regarding relevance [10 studies in the USA] and moderate concerns regarding methodological limitations [3 studies with moderate concerns and 1 with serious concerns IIIaim, methodology, data analysis, ethical, reflexivity, research design recruitment, findings]. |
| 4 | Women were pressured or coerced to terminate a wanted pregnancy, or were subject to physical violence with the intent of ending the pregnancy. Experiencing coerced terminations was sometimes the critical turning point for women to recognise the extent of the abuse within the relationship | [3, 5, 32–35, 38, 42, 43, 50, 52, 53, 54, 56] | High confidence | No to very minor concerns regarding adequacy and coherence, minor concerns regarding methodological limitations [reflexivity, recruitment, aims, methodology, data analysis, ethics, research design] and regarding relevance [mostly conducted in USA] |
| 5 | Many women were coerced by their partners into continuing pregnancies they may have otherwise terminated, sometimes due to feelings of guilt or fear of violence. Some women who sought an abortion were denied access to abortion services by their partners, such as by withholding money, denying transportation, and sabotaging appointments | [3, 29, 34, 39, 43, 50, 56] | Moderate confidence | No to very minor concerns regarding coherence, minor concerns regarding methodological limitations, and moderate concerns regarding regarding relevance [3 high income countries] and adequacy [7 studies with relatively thick data] |
| 6 | Despite perpetrating reproductive coercion and abuse with the intention to promote pregnancy, many men conveyed contradictory perspectives once the woman became pregnant. This included denial that he was the biological father, kicking the woman out of home, refusing to acknowledge the pregnancy once it was confirmed, and forcing the partner to have an abortion. | [3, 5, 29, 34, 35, 56, 58] | Moderate confidence | No or very minor concerns regarding methodological limitations and coherence, moderate concerns regarding relevance [studies mostly conducted in UK and USA] and adequacy [8 studies with somewhat thin data] |
| *Consequences of reproductive coercion and abuse* | | | | |
| 7 | Many women felt a reduced agency over their lives and reproductive outcomes as a consequence of the power imbalance in their relationship. Women expressed they considered the main goal of their partner's reproductive coercion and abuse was to control their lives and remove their ability for independent decision-making or leaving the relationship | [3, 29, 32–36, 39, 41, 43, 47, 48, 50, 56, 59] | Moderate confidence | No to very minor concerns regarding coherence, minor concerns regarding adequacy [14 studies with somewhat thick data] and moderate concerns regarding relevance [studies mostly conducted in high income countries] and methodological limitations [ethics, reflexivity, recruitment, data analysis, research design, methodology, aim] |
| 8 | In response to experiencing reproductive coercion and abuse, women experienced distress, anger and trauma | [5, 34, 38] | Low confidence | No to very minor concerns regarding methodological limitations and coherence but serious concerns regarding relevance [only studies conducted in 1 region] and adequacy [3 studies with relatively thin data] |

*(Continued)*

**Table 1.** (Continued)

| # | Summary of review finding | Studies contributing to the review finding | CERQual assessment | Explanation of CERQual assessment |
|---|---|---|---|---|
| 9 | Some women trivialised, minimized or blamed themselves for the coercive reproductive behaviour they experienced and did not recognise themselves as victims or survivors | [3, 5, 33, 49] | Low confidence | Minor concerns regarding coherence, methodological limitations, and relevance, and serious concerns regarding adequacy [5 studies with somewhat thin data] |
| *Women's responses to reproductive coercion and abuse* | | | | |
| 10 | A common mode of resistance to reproductive coercion and abuse was covert contraceptive use, which allowed a woman to exercise her reproductive autonomy | [3, 29, 30, 32, 33, 35–37, 42, 44–49, 51] | Moderate confidence | No to very minor concerns regarding coherence and adequacy; minor concerns regarding relevance [mostly studies conducted in USA and Kenya], and moderate concerns regarding methodological limitations [reflexivity, ethics, aims, methodology, data analysis] |
| 11 | Some women relied on peer support or actively sought help from female family members and community services to access or maintain contraception use and protect their safety while facing reproductive coercion and abuse | [42, 44] | Low confidence | No to very minor concerns regarding coherence and methodological limitations [reflexivity]; moderate concerns regarding relevance and serious concerns regarding adequacy [2 studies with somewhat thick data] |
| *Societal and cultural factors* | | | | |
| 12 | In some contexts, reproductive coercion and abuse was perpetrated by other family members and in-laws, where women's pregnancy intentions often differed from the familys' reproductive intentions. | [33, 38, 43, 44, 46, 50–52, 55] | Moderate confidence | No to very minor concerns regarding coherence, minor concerns regarding relevance and adequacy [10 studies with thick data] and moderate concerns regarding methodological limitations [reflexivity, aims, design, recruitment, methodology, data analysis, ethics] |
| 13 | Strong son preferences may result in reproductive coercion and abuse in some contexts. Such as sex selective abortions to terminate female foetuses, and if the firstborn child was female there may be increased pressure by husbands and in-laws to have closely spaced pregnancies. | [38, 44, 51, 52] | Moderate confidence | Minor concerns about relevance [may be more relevant for certain contexts with strong son preferences] and coherence and moderate concerns regarding methodological limitations [methodology, ethics, data analysis, reflexivity, research design] and adequacy [4 studies with somewhat thin data] |
| 14 | Women from African American communities acknowledged that systemic social inequities contributed to reproductive coercion and abuse, such as impending incarceration of male partners and barriers to housing and employment. They reported that these factors motivated men to use pregnancy coercion in order to form secure connections with female partners | [3, 29] | Low confidence | No to very minor concerns regarding coherence; minor concerns regarding methodological limitations [ethics, reflexivity, methodology, aim]; and serious concerns regarding relevance [only studies conducted in USA] and adequacy [2 studies with moderately thick data] |
| 15 | Some women from migrant backgrounds in Australia and the USA experienced the weaponizing of their visa status and threats of deportation if they did not comply with their partners reproductive coercion and abuse | [38, 42, 48] | Low confidence | No to very minor concerns regarding coherence and moderate concerns regarding relevance, adequacy III3 studies with somewhat thick data] and moderate concerns regarding methodological limitations IIIreflexivity, ethics, methodology, data analysis] |
| 16 | For some women, strictly defined gender roles placed direct pressure on women's 'biological imperative' to reproduce, and enabled the perpetration of reproductive coercion and abuse | [48, 51] | Low confidence | Minor concerns regarding coherence [findings are similar but not clear]; moderate concerns regarding relevance [only 2 countries] and serious concerns regarding methodological limitations [methodology, reflexivity, data analysis, ethics, research design] and adequacy [2 studies with thin data] |

**sabotage included disposing of, withholding, or blocking access to contraception, interfering with condom durability and condom refusal as well as psychological sabotage in the form of deceit regarding male infertility, misinformation and gaslighting to interfere with contraception (high confidence) [3, 29–31, 33–37, 42, 44, 47–49, 56–59].**

Partners interfered with condom durability by poking holes in the condom [35, 37, 58], and hid or disposed of contraception, for example by flushing pills down the toilet [3, 29, 30, 32, 33, 35–37, 42, 44, 47, 48]. While some partners used physical or verbal abuse to overtly sabotage contraceptive use [56, 57], some forms were more subtle, such as pressuring a woman to get an implant removed to have their partner's baby [42, 44, 49]. Contraceptive sabotage also happened to women using more discreet methods of contraception, such as a woman who

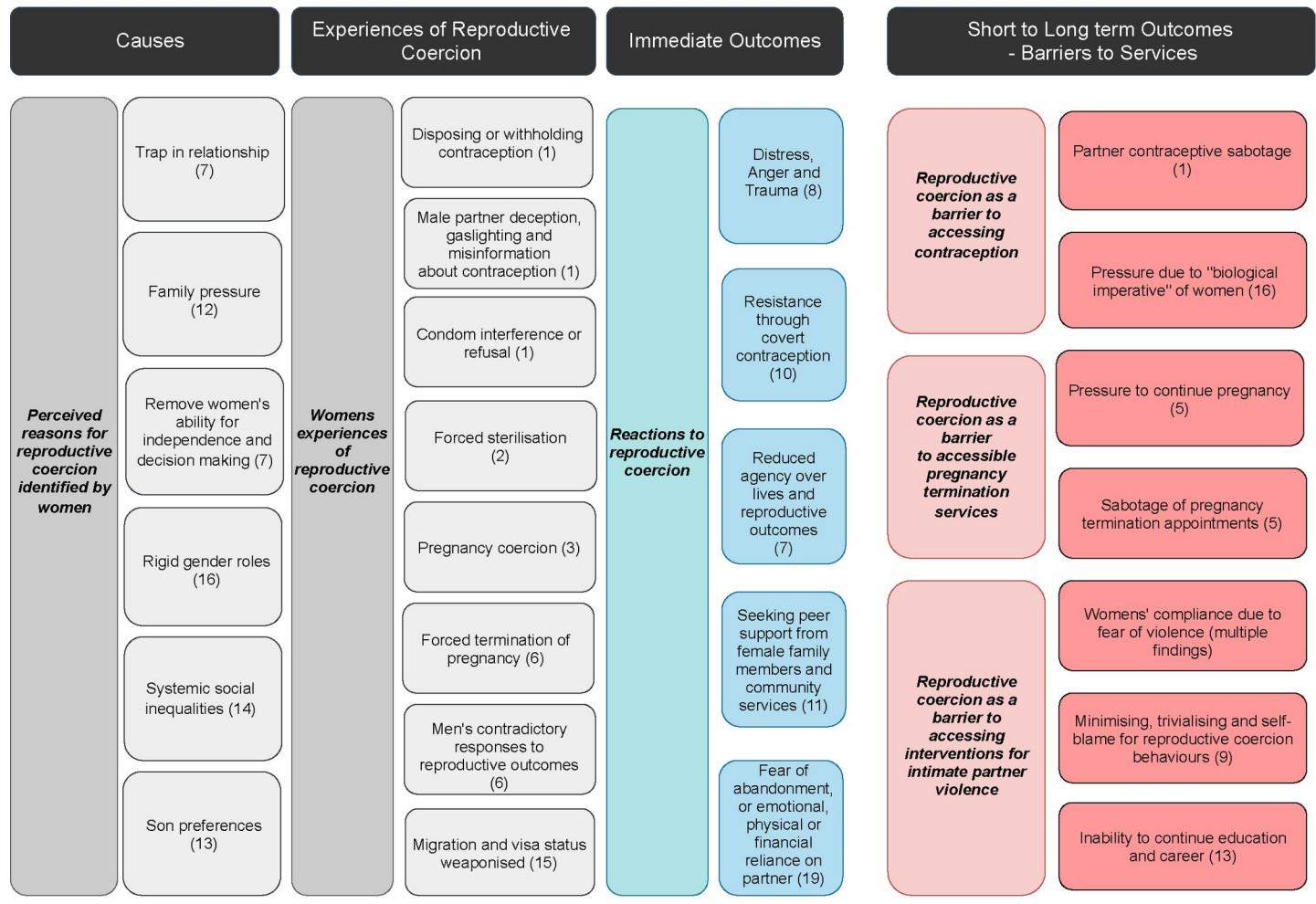

**Fig 2. Reproductive coercion and abuse: From causes to consequences.** This figure depicts the organisation of the review findings using a logic model approach. Moving from left to right, we identify the reasons for reproductive coercion and abuse identified by women, women's experiences of reproductive coercion and abuse, their reactions to reproductive coercion and abuse, and the short- and long-term outcomes of these experiences. Numbers in parenthesis show the connection to the review finding number.

used a NuvaRing after her partner prohibited her from using the pill. Her partner pulled NuvaRing out of her upon its discovery [3]. Partners also blocked access to contraception by assaulting them, or through other coercive means, such as ensuring women missed their family planning appointments [36, 37]. In one studies, a woman relied on their partner financially, but the partners refused to pay for contraception and the women were therefore unable to access it [3].

Male partner deceptions involved partners' lying about taking medicine that made them sterile [34], lying about having an operation [33] or claiming that they were unable to have children [42]. Partners' also attempted to dissuade women from using contraception by exaggerating negative side effects, such as that the contraceptive pill would lead to infertility, that it could cause hair loss and even death [3]. Some women consequently refrained from using contraception, fearing the damaging effects on their health [3]. Partners also used manipulation and 'gaslighting' to prevent women from taking the contraceptive pill. For example, a partner telling a woman that she had already taken her contraceptive pill when she had not, was a way of exerting control [56].

Partners also used condom refusal as a way of asserting their reproductive intentions. Tactics included the partner stating his displeasure in, and preference against, using a condom [29, 35, 48, 58, 59]. Men also refused more subtly, with one woman noting that her partner would 'disregard her' and not respond when she requested a condom, and continue with physical sexual advances [40]. Partners also used emotional manipulation if women requested condom use, such as accusing them of not trusting the relationship or cheating on him [3, 29], with the woman fearing relationship breakdown if she did not comply with his wishes [29]. While many women in the studies identified condom refusal and non-consensual removal in conjunction with other behaviours consistent with reproductive coercion and abuse, it is important to note that these forms of sexual violence can occur in isolation of intention to conceive [50]. Some studies noted additional contraceptive sabotage behaviours, however as it was unclear whether the intent was to cause pregnancy, these data have not been included in this finding. We also did not consider condom refusal to be reproductive coercion and abuse when women explicitly stated it was due to sexual pleasure concerns.

**Finding 2**

**Some women were forced by their partners into permanent methods of contraception, such as tubal ligation (low confidence) [33, 34].** This included a partner forcing a woman to have a tubal ligation, which caused her significant grief [33]. Another woman's partner convinced her to have a tubal ligation during their relationship break up, and she believed his intention was to ensure she was unable to have children with another man [34].

**Finding 3**

**Partners used a range of verbally and emotionally coercive behaviours, such as harassment, pressure and bullying, to promote pregnancies that were unwanted by women. Partners used emotional manipulation, including threatening to end the relationship, if women did not get pregnant, or telling family and friends that they were starting a family without her knowledge. Some women experienced sexual violence by their partners, with the intention to cause pregnancy, including rape and non-consensual unprotected sex (high confidence) [3, 29, 32–36, 38, 40, 41, 44, 48, 49, 51, 52, 56, 57].** This manipulation made women feel 'bullied' [29] with 'undue pressure' placed upon them to have a baby before feeling ready [40, 41, 44, 56, 57]. For example, after throwing away her contraception, a partner bought ovulation kits and four pregnancy tests to pressure her to conceive [29]. Among married couples in rural India, husbands exerted pressure on women to conceive early in marriage, have large families, and conceive sons [51]. Women were harassed by their husbands if unable to conceive and felt that their desire for children was misaligned with their partners, but had no option but to acquiesce to their partner's demands [40, 51]. Women also reported being forced to have sex against their will in order to promote pregnancy [3, 34, 36]. One woman reported that her partner sexually abused her while unconscious with the intent of pregnancy [36].

**Finding 4**

**Women were pressured or coerced to terminate a wanted pregnancy, or were subject to physical violence with the intent of ending the pregnancy. Experiencing coerced terminations was sometimes the critical turning point for women to recognise the extent of the abuse within the relationship (high confidence) [3, 5, 32–35, 38, 42, 43, 50, 52–54, 56].** This included threats of violence including threats to kill the woman and the baby if she did not terminate the pregnancy [34], and actual use of violence to end the pregnancy [3, 33, 34, 38, 42, 43, 50], with women reporting being thrown down the stairs or their partners punching or 'kicking the baby out' of the woman [3, 33, 34, 42]. Women were also emotionally coerced to terminate a pregnancy, with partners being unsupportive, threatening to end the relationship, accusing women of infidelity, 'wishing she would miscarry' or that she 'should go and get an abortion', causing women significant confusion in decision-making regarding pregnancy

termination [50, 54, 56]. For some women, violent attempts by partners to bring about a spontaneous abortion were successful [5, 42, 50]. Being forced to terminate a wanted pregnancy was a turning point for many women in recognising the extent of violence that was being used against them, with one participant describing the severe distress a 'watershed' moment [5].

**Finding 5**

**Many women were coerced by their partners into continuing pregnancies they may have otherwise terminated, sometimes due to feelings of guilt or fear of violence. Some women who sought an abortion were denied access to abortion services by their partners, such as by withholding money, denying transportation, and sabotaging appointments (moderate confidence) [3, 29, 34, 39, 43, 50, 56].**

Partners employed behaviours such as begging, badgering, promising to support the child, and guilt inducing comments such as 'You can't just kill your child' [3] or "I'll never forgive you" [3, 35, 50]. Others used subtler techniques to ensure their partners kept the pregnancy, including verbal pressure [29], as well as advising family and friends of the pregnancy without consulting her, thus compelling her to keep the baby to maintain the status quo [56]. Partners also sabotaged the appointments by making the woman eat so she was ineligible for general anaesthetic and causing a scene at the clinic which forced the woman to leave [3].

**Finding 6**

**Despite perpetrating reproductive coercion and abuse with the intention to promote pregnancy, many men conveyed contradictory perspectives once his partner became pregnant. This included denial that he was the biological father, kicking the woman out of home, refusing to acknowledge the pregnancy once it was confirmed, and forcing his partner to have an abortion (moderate confidence) [3, 5, 29, 34, 35, 40, 56, 58].** In a number of studies, men who had either pressured their partner to get pregnant or had thwarted measures to prevent pregnancy in the first place, then demanded their partner terminate the pregnancy [3, 5, 34, 35]. Men's behaviours ranged from purposefully getting their partner pregnant [35], prohibiting or sabotaging contraception [3, 34, 35], to exhibiting a careless attitude to unprotected sex using the 'new year' as an excuse to let 'whatever happens, happens' [5]. Once a pregnancy occurred, some men then exhibited controlling behaviour in contradiction with their previous stance - for example, a man forced his partner to have a termination and then referred to her as 'baby killer' [34], and a man hit his partner when he found out she was pregnant [58]. One participant stated that while she had not experience reproductive coercion and abuse herself, she acknowledged that many men are coercive to women, and that 'they get girls pregnant on purpose, just to leave them' [40].

## 3.2 | Consequences of reproductive coercion and abuse

This section identifies the consquences of reproductive coercion and abuse as perceived by women, including the partner's control over women, experiences of distress and trauma and women's minimisation of their reproductive coercion and abuse experiences. 18 studies explored these consequences [3, 5, 29, 32–36, 38, 39, 41, 43, 47–50, 56, 59].

**Finding 7**

**Many women felt a lack of reproductive control, and reduced agency over their reproductive outcomes as a consequence of the power imbalance in their relationship. Women expressed they considered the main goal of their partner's reproductive coercion and abuse was to control their lives and remove their ability for independent decision-making (moderate confidence) [3, 29, 32–36, 39, 41, 43, 47, 48, 50, 56, 59].**

Control was an overarching theme identified by women when reflecting on their experiences of reproductive coercion and abuse [29, 32–34, 43, 48, 50, 56]. Many women who were

subjected to reproductive coercion and abuse were also subjected to control in other aspects of their lives [29, 32, 48, 56]. This included limiting freedom of movement such as using death threats or locking a woman inside her home to prevent her from leaving, prohibiting her from driving a car, monitoring phone calls and her eating habits, controlling finances, controlling internet access as well as covertly recording her at home [29, 32, 48, 56]. Some male partners spoke of wanting to have a baby to stop women from leaving [3, 36, 39]. Some women believed they were unable to make decisions as they did not have the power to do so [32]. Perceptions of reduced agency were also acknowledged, with one stating that pregnancy cannot be planned for, and that 'lots of people don't want to get pregnant but it just happens' [30]. Some women had their ability to get pregnant "weaponised against them" as a form of control [43]. And, in one study, the children that resulted from reproductive coercion and abuse were used to assort further control over the life of a woman [50]. Women also believed that reproductive coercion and abuse enabled their partners to trap them in the relationship [3, 32, 33, 36, 39, 48]. For example, women explained that having children 'makes you need them [partners]' [32], that men wanted pregnancy to 'keep them down' [33], and that men were 'scared that [they] would run away' unless they became pregnant [48]. Many women felt that their partner desired a child as a way of controlling them [32], even likening having a child to using 'guns to control' [32] a relationship. Echoing women's beliefs, male partners spoke of wanting to have a baby to 'know that she will be in my life forever' [3] and to try to get her pregnant 'so she wouldn't leave him'[36]. Some women felt they were unable to resist their partner's reproductive coercion and abuse, despite women's opposition to starting a family due to desires to continue education or follow career goals [3, 29, 35, 41, 59].

**Finding 8**

**In response to experiencing reproductive coercion and abuse, women experienced distress, anger and trauma (low confidence) [5, 34, 38].**

Women who were forced to have an abortion experienced severe trauma, distress and depressive symptoms [5, 34, 38], with one woman experiencing suicidal ideation after her partner threatened to kill her and their child if she did not go ahead with the abortion [34]. For women who experienced non-consensual condom removal, fear of sexually transmitted infections and pregnancy, coupled with shame, anxiety, anger, insecurity and confusion were common experiences [5].

**Finding 9**

**Some women trivialised, minimized or blamed themselves for the coercive reproductive behaviour they experienced and did not recognise themselves as victims or survivors (low confidence) [3, 5, 33, 40, 49, 56].**

Women may laugh off requests to stop contraceptive use [49], or consider non-consensual condom removal as 'pathetic,' 'trivial' or a joke on the partner's behalf [5]. These women were concerned that if they identified as a victim, they would have to address their partner and his behaviour; thus trivialising the coercion relieved them from confrontation [5]. Further, minimising the behaviour allowed women to suppress any feelings of distress or trauma, that as one woman described is like 'a weight off your shoulders' [5]. Women reported that love for their partner distorted their perception of violent behaviour [3, 5], and cause them to minimise the severity of the behaviour they were experiencing, with women playing down the severity of the partner's coercion [5]. Other reasons for minimising the importance of the coercive behaviours included failing to perceive it as violence, particularly in circumstances where other forms of intimate partner violence were present [5]. In some cases, women were manipulated by their partners to doubt the seriousness of the reproductive coercion and abuse they experienced [5].

These women attributed their lack of control over contraceptive use to their 'chaotic life-style', while some considered themselves 'weak' for allowing the pregnancy to occur [56]. Many took responsibility for the experience, believing they should have expressed more clearly that they had not consented to sex without a condom or that they should have spoken up sooner [5]. One participant acknowledged that the emotional abuse led her to feel that something was wrong with her because she did not match her partners desire to become pregnant [40]. Some also questioned whether they were to blame, lacking a clear recollection of events due to drinking alcohol at the time, and believing they 'may have consented to taking off the condom' [5].

### 3.3 | Women's responses to reproductive coercion and abuse

Women's responses to reproduction was explored in 16 studies [3, 29, 30, 32, 33, 35–37, 42, 44–49, 51]. This included the use of covert contraception to resist coercion and help-seeking.

**Finding 10**

**A common mode of resistance to reproductive coercion and abuse was covert contraceptive use, which allowed a woman to exercise her reproductive autonomy (moderate confidence) [3, 29, 30, 32, 33, 35–37, 42, 44–49, 51].**

For example, a woman had a secret compartment in her purse where she stored contraceptive pills [36], while other women covertly used contraceptive methods that were able to be hidden such as a transdermal hormonal patch [3], Depo-Provera [35, 44, 45, 47–49], contraceptive implant [42, 45] or an intrauterine device (IUD) [29, 48, 51]. Another woman had a tubal ligation following childbirth to ensure she would not be able to conceive again [33]. Some women also used deception to hide their covert use, use including: telling their partner that their contraceptive pills were vitamins, telling their partner they were menstruating to avoid sex, or suggesting to their partner that there may be something wrong with his fertility if conception had not occurred after she received a contraception injection [42, 48]. Covert use may contribute to women's feelings of conflict and shame about disobeying their husbands but women expressed that covert contraception was a necessary last resort to avoid pregnancy [51]. Some women would take advantage of situations where their partners granted permission to leave the house, such as running errands or getting their children immunised, to access contraception [45]. Conversely, some women who were using contraception covertly discontinued use after their partners found out [46].

**Finding 11**

**Some women relied on peer support or actively sought help from female family members and community services to access or maintain contraception use and protect their safety while facing reproductive coercion and abuse (low confidence) [42, 44].**

Women reported receiving or providing support for other women regarding contraception use such as: help hiding their contraceptive or appointment clinic cards, advice about less detectable contraceptive methods such as depo-provera injections, and sending emergency contraception [42, 44]. Community and social services such as saving clubs, children's schools and health clinics were reported by some women as safe places to receive help regarding contraception [42, 44].

### 3.4 | Societal and cultural factors

Thirteen studies explored different societal and cultural aspects of women's experiences of reproductive coercion and abuse [3, 29, 33, 38, 42–44, 46, 48, 50–52, 55]. This includes perpetration of reproductive coercion and abuse and abuse by family and in-laws, systemic

inequalities, the preference of partners and their families for sons and rigid gender roles in which women are pressured to conceive.

**Finding 12**

**In some contexts, reproductive coercion and abuse was perpetrated by other family members and in-laws, where women's pregnancy intentions differed from the familys' reproductive intentions (moderate confidence) [33, 38, 43, 44, 46, 50–52, 55].**

Family reproductive control was experienced by women from five different countries. Four studies were from the USA, with one study of low income African American and White women [33], two studies of mainly low income African American women [40, 43], and one study of Indian migrants to the USA [38]. Two other studies described reproductive coercion and abuse perpetrated by family members in India [51, 52], with Indian women's bodies described as having 'many stakeholders' and reproductive decisions being made collectively with women often under duress [52]. Two further studies were from Kenya [44, 46],one study from Iran [55] and another from Australia [50]. Many women reported families that were both physically and verbally abusive, especially if the family's expectations of the women's childbearing were not met [33, 44, 51, 55]. Mothers-in-law and partners were the biggest sources of pressure to conceive, and mothers-in-law commonly controlled the outcome of the pregnancy, both for pregnancy continuation or termination [38, 50–52]. As an example, one woman had not consented to an abortion, however her mother-in-law took her to a public hospital where the abortion was performed with no anaesthesia [52]. Some women reported in-laws as a direct barrier to contraception, such as having in-laws "who threatened to report them to their partners" when using covert contraception [46]. The sister in-law of another woman refused to support her with childcare to stop her from getting a contraceptive injection [43]. In one study of low-income African American women, participants perceived pregnancy pressure from in-laws, with their partner's mother intensifying their son's pregnancy coercion [40]. Other participants perceived that this pressure was not ill-intentioned, but the desires of the family were incongruent to their own reproductive intentions [40].

**Finding 13**

**Strong son preferences may result in reproductive coercion and abuse in some contexts, such as sex selective abortions to terminate female foetuses. If the firstborn child was female there may be increased pressure by husbands and in-laws to have closely spaced pregnancies (moderate confidence) [38, 44, 51, 52].**

This was particularly relevant in studies of Indian communities. Some culturally-specific forms of shaming towards Indian women in the USA including threatening to send those unable to conceive sons back to their families in India, or threatening that their son would leave her for a woman able to produce grandsons [38]. It should be noted that son preference may be changing over time, with only one woman stating that a son is necessary in a recent study of rural Hindu women [52]. In one study from Kenya, some women reported that reproductive coercion and abuse reduced or stopped completely once their partners or in-laws considered that the woman had enough male children [44].

**Finding 14**

**Women from African American communities acknowledged that systemic social inequities contributed to reproductive coercion and abuse, such as impending incarceration of male partners and barriers to housing and employment. They reported that these factors motivated men to use pregnancy coercion in order to form secure connections with female partners (low confidence) [3, 29].** With knowledge of their impending incarceration, some men in these communities coerced their partners to become pregnant to reduce the chances of the woman leaving the relationship, believing the pregnancy would make her less desirable to other men [3]. One woman considered her partner's intention was to ensure that she would be

more invested in maintaining a relationship with him, and to secure a lifetime of social, economic and housing support upon his release by virtue of being the father of her child [29].

**Finding 15**

**Some women from migrant backgrounds in Australia and the USA experienced the weaponizing of their visa status and threats of deportation if they did not comply with their partners reproductive coercion and abuse (low confidence) [38, 42, 48].**

Women had grave fears of abandonment or divorce if they did not conceive a son or if they did not have family in the USA and feared being sent back to India [38]. In Australia, migrant women experienced increased risk of reproductive coercion and abuse due to their partners using threats of deportation or using their citizenship status as leverage against them, and women being geographically distant from their usual support systems [42, 48].

**Finding 16**

**For some women, strictly defined gender roles placed direct pressure on women's 'biological imperative' to reproduce and enabled the perpetration of reproductive coercion and abuse (low confidence) [40, 48, 51].** These roles included the expectation on women to produce children shortly after marriage, be sexually available to their husbands [51], have babies because it is 'all natural' [48], or that women are 'useless' if unable to have their partner's child [40].

## 4 | Discussion

This qualitative evidence synthesis presents women's experiences of, responses to, and consequences of reproductive coercion and abuse and abuse. Women shared experiences of pregnancy coercion, contraceptive sabotage or control of pregnancy outcomes. Women responded to reproductive coercion and abuse through resistance such as covert contraceptive use, and help-seeking. The impacts of women's experiences of reproductive coercion and abuse included trauma, unintended pregnancy, and inability to access contraception and abortion services when required. Women often did not access interventions for intimate partner violence.

Our qualitative findings demonstrate behaviours describd previously in prevalence literature [10, 60] related to contraceptive control [7, 11, 60–62], pregnancy coercion [2, 7, 60, 63] and control of pregnancy outcome [2, 7, 64, 65]. Our qualitative evidence synthesis expands on prevalence estimates by exploring how, and in what context, these behaviours manifest, as well as the discourse surrounding the experience. Likewise, resistance strategies, such as covert contraceptive use, have been reported in quantitative research [7]. For example, a prevalence study on use of injectable methods of contraception found that women who experienced reproductive coercion and abuse had higher rates of use than other methods [62]. Our qualitative findings contribute to understanding how women might resist reproductive coercion and abuse, which can assist early response and intervention development.

A systematic review found that women experiencing intimate partner violence were less likely to use condoms and oral contraceptives [66]. We also found that reproductive coercion and abuse was a major barrier to contraceptive choice and use, and limited a woman's sense of self efficacy due to partner control and threat of violence. Reproductive coercion and abuse has also been determined as a barrier to termination services in previous studies [65, 67] due to geographical access to the clinic and difficulty in obtaining a surgical termination secretly due to easy detection [65].

Our study has both limitations and strengths. It is possible that due to evolving language and scope of what is understood as reproductive coercion and abuse, some key relevant studies may not have been included. Based on the languages of the review authors, we were only able

to include studies published in English or Spanish. Six studies are classified as 'awaiting classification' due to publication in another language and were not included in the analysis. We explored cis-women's experiences of reproductive coercion and abuse, but note that cis-men, non-binary, gender-diverse, and trans-people may also experience reproductive coercion and abuse, which could be explored in future research. Another limitation is that women's use of covert contraception may not necessarily be due to their partner's desire for pregnancy but a range of other factors. This includes male sexual pleasure and dislike of side effects including weight gain and increased and irregular bleeding.

There are a number of key strengths to this review. The use of the Cochrane systematic review methodology was a strength of the study. Due to the relatively recent coining of 'reproductive coercion and abuse' as a unique phenomenon, a broad search strategy enabled us to capture experiences of behaviours defined under different terminology including reproductive control, and the intersection of reproduction, unintended pregnancy and intimate partner violence. Another key strength was that we only included data describing the types of behaviours that perpetrators use when there was clear intent of a reproductive outcome, particularly pertaining to contraceptive sabotage.

We also identified opportunities for future research and implications for policy and practice. First, while our review was global, most studies were conducted in the USA. More research is needed on women's experiences of reproductive coercion and abuse in other settings, as the manifestations, perpetrators and outcomes may differ. Likewise, studies of migrant women suggested that these women may experience heightened risk to reproductive coercion and abuse due to their immigration status, which sometimes also interfered with their ability to seek help [38, 42, 48]. There is a particular need for more research on the intersection of reproductive coercion and abuse and faith, with little being known about reproductive coercion and abuse experienced by women of faith, perpetrated by men of faith, and in the context of interfaith relationships. Second, few studies explore the perpetrator's perspectives [7]. Research in this area will aid in the understanding of why individuals engage in reproductive coercion and abuse and the risk factors for perpetration, which will better inform which perpetrator-focused interventions may be most effective. Third, there is an association between visits to reproductive health services and experiences of reproductive coercion and abuse [7]. Health professionals should ensure that best practice screening methods are used to increase detection and early intervention of reproductive coercion and abuse. Further research on help seeking and appropriate interventions should be conducted to help inform practice and ensure that women experiencing reproductive coercion and abuse are detected and provided appropriate support and service referrals.

For women experiencing reproductive coercion and abuse, services need to be adapted to ensure that women can safely access contraception and termination when required.

To minimise harm, a women's circumstances and needs should be assessed, with appropriate and safe contraceptive options suggested accordingly. This may include forms of contraception that can be covert (IUDs, injectables, or surgical sterilisation) as well as the option of same day insertion to minimise the number of appointments required. Staff at termination services need to be provided with appropriate training on the barriers women experiencing reproductive coercion and abuse may face when trying to procure a termination of pregnancy [68]. Appropriate screening is also required prior to a termination to ensure women give their informed consent, and to ensure women's safety during and following the procedure. Community services could also be a useful setting for women to seek help safely. Increasing access to early medical terminations through legislative and practice changes may aid women experiencing reproductive coercion and abuse to more readily access termination services.

## 5 | Conclusions

Reproductive coercion and abuse is a hidden and emergent public health issue. This qualitative evidence synthesis explored the experiences and perceptions of a range of women and found common patterns of the different manifestations of reproductive coercion and abuse, reasons for reproductive coercion and abuse identified by women, women's reactions to reproductive coercion and abuse as well as barriers women face to accessing appropriate contraception, pregnancy termination services and interventions for intimate partner violence. This review reinforced the importance of socio-cultural factors in understanding the phenomenon of reproductive coercion and abuse and how it affects women, as well as how the mechanisms of power and control at both individual and societal levels work to perpetuate the incidence of reproductive coercion and abuse against women.

## Supporting information

**S1 Checklist.**
(DOCX)

**S1 Table. ENTREQ statement.** Completed ENTREQ statement for enhancing transparency in reporting the synthesis of qualitative research.
(DOCX)

**S2 Table. Search strategies.** Search strategies for MEDLINE, Embase, and CINAHL databases.
(DOCX)

**S3 Table. Critical appraisal.** Critical appraisal of included studies using the CASP tool.
(DOCX)

**S4 Table. Characteristics of included studies.** Characteristics of included studies.
(DOCX)

**S5 Table. Codebook.** Qualitative codebook developed during analysis.
(DOCX)

**S1 File.**
(DOCX)

## Acknowledgments

We thank our colleague Nilab Hamidi who provided assistance with screening titles and abstracts in the early stages of this project.

## Authors' information

**Review author reflexivity.** Our review team has expertise in social sciences, sexual and reproductive health , social psychology and public health. From the onset of the study, we believed that reproductive coercion and abuse is a form of violence against women and a method that is used to exert control over women. We consider reproductive coercion and abuse to be a major public health issue with major negative impacts on the women who experience it. We identify as women and have professional experience in women's health, women's rights advocacy, qualitative research, and gender theory. We are aware our perceptions around reproductive coercion and abuse could have an influence in the methodology of the research, including study design, search strategy, inclusion of articles, data extraction, analysis and synthesis as

well as interpretation of the findings. Therefore, through the research process we had continuous conversations with each other exploring our views on reproductive coercion and abuse and the findings, while holding each other accountable to stay truthful to the research principles of honesty, rigour and transparency.

## Author Contributions

**Conceptualization:** Cathy Vaughan, Meghan A. Bohren.

**Data curation:** Martha Isela Vazquez Corona.

**Formal analysis:** Jessica E. Moulton, Martha Isela Vazquez Corona, Meghan A. Bohren.

**Methodology:** Jessica E. Moulton, Meghan A. Bohren.

**Supervision:** Cathy Vaughan, Meghan A. Bohren.

**Visualization:** Jessica E. Moulton.

**Writing – original draft:** Jessica E. Moulton, Martha Isela Vazquez Corona.

**Writing – review & editing:** Jessica E. Moulton, Martha Isela Vazquez Corona, Cathy Vaughan, Meghan A. Bohren.

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
