## [Decision Letter · Decision Letter 0]

22 Sep 2021

PONE-D-21-21336Women’s perceptions and experiences of reproductive coercion and abuse: a qualitative evidence synthesisPLOS ONE

Dear Dr. Moulton,

Thank you for submitting your manuscript to PLOS ONE. After careful consideration, we feel that it has merit but does not fully meet PLOS ONE’s publication criteria as it currently stands. Therefore, we invite you to submit a revised version of the manuscript that addresses the points raised during the review process.

The two reviewers are experts in the field and have provided valuable feedback. Please respond to each of the criticisms articulated by the reviewers. If a reviewer asks you to do something that you cannot do, please be thorough in your explanation of why you cannot comply.

We look forward to receiving your revised manuscript.

Kind regards,

Alison Gemmill

Academic Editor

PLOS ONE

Journal Requirements:

2. We note that this manuscript is a systematic review or meta-analysis; our author guidelines therefore require that you use PRISMA guidance to help improve reporting quality of this type of study. Please upload copies of the completed PRISMA checklist as Supporting Information with a file name “PRISMA checklist

Reviewers' comments:

Reviewer's Responses to Questions

**Comments to the Author**

1. Is the manuscript technically sound, and do the data support the conclusions?

Reviewer #1: Yes

Reviewer #2: Yes

2. Has the statistical analysis been performed appropriately and rigorously? 

Reviewer #1: N/A

Reviewer #2: N/A

3. Have the authors made all data underlying the findings in their manuscript fully available?

Reviewer #1: Yes

Reviewer #2: Yes

4. Is the manuscript presented in an intelligible fashion and written in standard English?

Reviewer #1: Yes

Reviewer #2: Yes

5. Review Comments to the Author

Reviewer #1: Thank you for the opportunity to review this manuscript, which is a synthesis of qualitative data on the topic of RC. This review is rigorously conducted and well-written. My main concern is that an exhaustive and comprehensive review such as this is better suited for an academic paper (as it states it was originally written) than a manuscript. I think it can be scaled back and more narrowly focused to be more appropriate for publication. Findings need to be further synthesized for readability and to avoid presenting 15 pages of data. Scope could be narrowed to focus on women’s experiences OR women’s reactions OR women’s perceptions of reasons for RC. The theoretical framework (section 2.1) and methods sections on lines 254-295 can be significantly scaled back. The Table is helpful but could be more helpful – needs to summarize topics more broadly and succinctly to be useful as a table. I am also wondering if the study was limited to the actual qualitative data that was presented in each study? In my study, which was included in the review, I sometimes describe a range of behaviors that emerged in interviews, and then present a quote to illustrate the theme, but my study was only cited in this review regarding the behavior in the one quote. That seems to be a limitation that should be acknowledged and more intentionally addressed.

Specifics:

Line 129: Citation 8 is also published in 2010 – would cite earlier literature here to support the point being made.

Line 132: some studies find greater than 16% prevalence, for example:

• Alexander, K. A., Willie, T. C., McDonald-Mosley, R., Campbell, J. C., Miller, E., & Decker, M. R. (2019). Associations Between Reproductive Coercion, Partner Violence, and Mental Health Symptoms Among Young Black Women in Baltimore, Maryland. Journal of Interpersonal Violence, 088626051986090. https://doi.org/10.1177/0886260519860900

• Grace KT, Perrin N, Clough A, Miller E, Glass NE. Correlates of reproductive coercion among college women in abusive relationships: Baseline data from the College Safety Study. J Am Coll Heal 2020;0:1–8. https://doi.org/10.1080/07448481.2020.1790570.

• Holliday CN, McCauley HL, Silverman JG, et al. Racial/Ethnic Differences in Women’s Experiences of Reproductive Coercion, Intimate Partner Violence, and Unintended Pregnancy. J Women’s Heal. 2017;26(8):1-8. doi:10.1089/jwh.2016.5996

Lines 224-227 seem to repeat the paragraph above?

Lines 308-309: “All women in the studies were in relationships with male partners at the time they experienced reproductive coercion.” I think it’s more accurate to say all women experienced RC from male partners. Can’t necessarily say they were in a relationship.

Lines 514-515: are studies of men included in this review? Some studies are absent if so (Alexander, K. A., Sanders, R. A., Grace, K. T., Thorpe, R. J., Doro, E., & Bowleg, L. (2019). “Having a Child Meant I Had a Real Life”: Reproductive Coercion and Childbearing Motivations among Young Black Men Living in Baltimore. Journal of Interpersonal Violence, 1–29. https://doi.org/10.1177/0886260519853400)

Study Flow Diagram: “reports not retrieved” – should this say “unable to be retrieved”? Or is there another reason they were not retrieved? What does “wrong setting” and “wrong population” refer to? What is the difference between the 2 categories in the “Included” section?

Logic model: this is really helpful in synthesizing the large amount of data in this paper. Some comments: Accusations of infidelity is a tactic, not a cause of RC. I am not seeing the cultural piece among causes? Numbers need to be explained. “Women’s partners reaction to resistance” should really move one column to the right, since this is referring to a reaction to an immediate outcome. Section on “Short to Long Term Outcomes – Barriers to Services” is less clear to me as a distinct category. I think it could be eliminated altogether or else the scope of the paper could be narrowed to “Outcomes”, and then this section could be fleshed out and organized a bit more.

Some minutiae:

Line 224 “constitute”

Line 402: suggest word other than “heartbreak”

Line 468: “experienced”

Line 882: suggest “may” instead of “will

Reviewer #2: Thank-you for the opportunity to review this manuscript. The article presents the results of a systematic review and meta-analysis of qualitative studies exploring the lived experience of reproductive coercion and abuse. I commend the authors on an excellently-conducted, well-presented, timely and topical review that will make a fantastic contribution to the literature. I particularly appreciate the extra efforts to include not only studies explicitly addressing reproductive coercion, but also earlier studies describing behaviours that meet the criteria for RC but which are not labeled this way by researchers or participants. It is a pity that the review was not registered with PROSPERO or other similar platform, however, the methodology is clearly-described and sound, and the team are experienced in undertaking these types of reviews, so I think it is highly unlikely to be an issue.

I have some suggestions below to improve the manuscript which I hope the authors will find helpful:

1. I would strongly suggest that the authors critically engage with the work of Tarzia and Hegarty (2021). Their conceptual work around RC would assist in both interpreting the findings of this review and in ensuring that there is clarity around the behaviours being described.

2. Terminology - although recent research has opted for the terms "reproductive coercion and abuse" or "reproductive abuse" the authors have chosen to stick to the older term "reproductive coercion". This is absolutely fine but I would like to see a justification for why they have done this. I assume it is to avoid confusion since most of the literature they are reviewing will use RC but it would be good to state this explicitly.

3. Line 132 - please note that the highest prevalence estimates are around 30%. Please see the recent systematic review by Rowland and Walker for a summary of prevalence rates across different settings.

4. Line 141 the authors outline some of the background literature on risk factors and prevalence. A sentence should be added here qualifying that measurement issues render some of this work problematic given that different behaviours are included across different studies and many studies are either over or under-inclusive. The Tarzia & Hegarty 2021 paper outlines this issue.

5. Line 149 should have a citation. Suggest Tarzia L, Douglas H, Sheeran N. Reproductive coercion and abuse against women from minority ethnic backgrounds: views of service providers in Australia. Cult Health Sex. 2021 Jan 11:1-28.

6. Line 176 - The authors state that the review takes a phenomenological approach, however, the findings do not align with this. Simply reporting types of behaviours and women's responses to them does not describe the meanings participants place on their experiences. Not everything that examines people's experiences in a qualitative way is "phenomenological". Please reconsider whether another approach describes your analysis more accurately.

7. Line 181 - Similar to my previous point, the authors claim to have drawn on a number of frameworks to help them interpret the findings, without insufficient critical engagement with what these frameworks actually do. For example, acknowledging gendered power relations in society is just... feminism? Connell's gender regime (from what I understand) is a particular way of describing gendered structural relations within "an organisation". If you are simply talking about broader gender inequality in society then I don't think Connell's theory is really what you need. To be clear I am not disagreeing with your argument that social structures condition men's and women's social roles, just that Connell's theory is being mis-applied or not adequately explained. Similarly, the claim that the "ecological framework" has been used to interpret the findings needs further explanation or justification. If all you are doing is acknowledging the cultural specificity of women's experiences of RC then this is not "ecological". The ecological framework does a very specific thing which is look at VAW at all levels (individual, relationship, community, society) and you have not really done that in interpreting your findings here. Neither is there any evidence (contrary to the claims in Line 191 onwards) that the ecological factors identified by Heise in relation to VAW more broadly are actually applicable to the RC context. To be clear I am not suggesting you rework your findings just that you avoid using fancy frameworks to describe what you did unless they are actually being used properly.

Lines 217-219 - The definition and measurement tool are problematic. Please justify why these definitions have been used as opposed to more recent definitions and frameworks (see Tarzia & Hegarty 2021). For example Miller's measurement tool completely excludes abortion coercion as a form of RC. Happily I don't think you actually *did* use the Miller measurement tool in your analysis since the rest of the paper is very specific about the intent of the behaviour so not really sure why this is here.

Line 261 please explain briefly how the CASP was adapted

Lines 366 onwards (Results section)

Do all of the studies describing contraceptive sabotage do so in the context of DELIBERATE pregnancy intention (as opposed to "stealthing")? It is fantastic to see the specific attention paid to this detail (intent) throughout the analysis, however, it is less clear whether the included studies have been curated with this exclusion in mind? For example, the quotes on lines 387-388 clearly do not relate to pregnancy intent but rather, to other motivations for removing a condom. Please clarify.

There is a broader issue of overlap and confusion within the Findings. There are some Findings that do not seem to fit well under their respective theme headings, and there are many areas of overlap. I would strongly recommend the authors review the Findings with the aim of consolidating some of them and re-assessing the appropriateness of the headings. For example, Finding 10 suggests that men perpetrate RC as a way of bolstering their ego. However, it's unclear how the men "showing off the baby" necessarily demonstrates reproductive coercion? Is it clear that in these studies, the women believed that men wanted a child to satisfy their egos - AND specifically coerced them into having one? Similarly, Finding 13 does not seem to be describing a motivation for perpetrating RC. Finding 15 includes a lot of extra content that describes the behaviours/mechanisms of RC but is not relevant to understanding WHY the perpetrator does it (which is the focus of the theme it is situated under). This issue of overlap is particularly problematic in the themes on Barriers. A lot of the content described under "Barriers to accessing IPV interventions" for instance is not actually about RC as a barrier to IPV interventions. The same goes for "Barriers to termination" and "Barriers to contraception". A lot of this content seems to sit better under the types of behaviours that perpetrators use.

The Discussion section needs some minor revisions relating to my comments above (regarding the definitional issues and use of frameworks).

I look forward to citing this review when it is published.

6. PLOS authors have the option to publish the peer review history of their article (what does this mean?). If published, this will include your full peer review and any attached files.

Reviewer #1: No

Reviewer #2: No

---

## [Author Response · Author response to Decision Letter 0]

16 Nov 2021

Dear Editors,

Thank you and the reviewers for your thoughtful consideration of our manuscript. We have revised based on the feedback in a tracked changes version of the manuscript and explained the revisions and responded to questions in table form, point-by-point. Please see attached response to reviewers letter.

Thank you for your consideration, we look forward to hearing from you.

Sincerely,

Jessica E Moulton, on behalf of the authors

---

## [Editor Report · Decision Letter 1]

6 Dec 2021

Women’s perceptions and experiences of reproductive coercion and abuse: a qualitative evidence synthesis

PONE-D-21-21336R1

Dear Dr. Moulton,

We’re pleased to inform you that your manuscript has been judged scientifically suitable for publication and will be formally accepted for publication once it meets all outstanding technical requirements.

Kind regards,

Alison Gemmill

Academic Editor

PLOS ONE

Additional Editor Comments (optional):

One minor comment-- please change the word "Caucasian" to "White"
---

## [Editor Report · Acceptance letter]

9 Dec 2021

PONE-D-21-21336R1 

Women’s perceptions and experiences of reproductive coercion and abuse: a qualitative evidence synthesis 

Dear Dr. Moulton:

I'm pleased to inform you that your manuscript has been deemed suitable for publication in PLOS ONE. Congratulations! Your manuscript is now with our production department. 

Kind regards, 

on behalf of

Dr. Alison Gemmill 

Academic Editor

PLOS ONE